# Ternary Blended Binder for Production of a Novel Type of Lightweight Repair Mortar

**DOI:** 10.3390/ma12060996

**Published:** 2019-03-26

**Authors:** Milena Pavlíková, Lucie Zemanová, Martina Záleská, Jaroslav Pokorný, Michal Lojka, Ondřej Jankovský, Zbyšek Pavlík

**Affiliations:** 1Department of Materials Engineering and Chemistry, Faculty of Civil Engineering, Czech Technical University in Prague, Thákurova 7, 166 29 Prague 6, Czech Republic; milena.pavlikova@fsv.cvut.cz (M.P.); lucie.zemanova@fsv.cvut.cz (L.Z.); martina.zaleska@fsv.cvut.cz (M.Z.); jaroslav.pokorny@fsv.cvut.cz (J.P.); 2Department of Inorganic Chemistry, Faculty of Chemical Technology, University of Chemistry and Technology, Technická 5, 166 28 Prague 6, Czech Republic; michal.lojka@vscht.cz (M.L.); ondrej.jankovsky@vscht.cz (O.J.)

**Keywords:** biomass combustion, ternary blended binder, lightweight mortar, pozzolanic activity, functional properties

## Abstract

The goal of the paper was development and testing of a novel type of ternary blended binder based on lime hydrate, metakaolin, and biomass ash that was studied as a binding material for production of lightweight mortar for renovation purposes. The biomass ash used as one of binder components was coming from wood chips ash combustion in a biomass heating plant. The raw ash was mechanically activated by grinding. In mortar composition, wood chips ash and metakaolin were used as partial substitutes of lime hydrate. Silica sand of particle size fraction 0–2 mm was mixed from three normalized sand fractions. For the evaluation of the effect of biomass ash and metakaolin incorporation in mortar mix on material properties, reference lime mortar was tested as well. Among the basic physical characterization of biomass ash, metakaolin and lime hydrate, specific density, specific surface, and particle size distribution were assessed. Their chemical composition was measured by X-Ray fluorescence analysis (XRF), morphology was examined using scanning electron microscopy (SEM), elements mapping was performed using energy dispersive spectroscopy (EDS) analyser, and mineralogical composition was tested using X-Ray diffraction (XRD). For the developed mortars, set of structural, mechanical, hygric, and thermal properties was assessed. The mortars with ternary blended binder exhibited improved mechanical resistance, lower thermal conductivity, and increased water vapor permeability compared to the reference lime mortar. Based on good functional performance of the produced mortar, the tested biomass ash could potentially represent a novel sustainable alternative to other pozzolans commonly used in construction industry. Moreover, reuse of biomass ash in production of building materials is highly beneficial both from the environmental and economic reasons especially taking into account circular economy principles. The ternary blended binder examined in this paper can find use in both rendering and walling repair mortars meeting the requirements of culture heritage authorities and technical standards.

## 1. Introduction

Non-hydraulic or sub-hydraulic mortars were used from ancient times until the early 19th century [1]. Nevertheless, first attempts to use hydraulic mortars have been already recognized in Greek and Roman periods [2]. From the half of 19th century, the natural and later synthetic hydraulic lime was used. After the first cement was discovered in the first half of 19th century, cement becomes prevailing binder of construction industry [2,3]. Starting from the beginning of 20th century, with the expansion of Portland cement (PC) production, lime plasters were replaced with cement plasters or cement-lime plaster. As lime was used for walling and rendering of buildings for centuries, lime-based plasters are one of the main components of a significant number of old buildings [4,5,6,7]. Presently, the application potential of lime mortars is reduced due to the intensive use of PC. On the other hand, cement-base materials possess low water vapor transmission rate [8,9]. Moreover, the use of hydraulic binders in repair mortars has been associated with compatibility problems due to their high strength, stiffness, and low vapor permeability [10,11,12]. It is generally accepted that the PC used in mortars for conservation and repair of historical buildings was a wrong choice, as renovated structures were often associated with the structural, hygrothermal, and other building pathology problems [13,14]. The culture heritage authorities put emphasis on the use of traditional building materials and techniques in the preservation of built heritage [15,16,17]. Therefore, lime-based mortars have been used for this purpose in order to achieve the required compatibility with the originally used materials [18,19]. Unfortunately, lime based mortars possess some disadvantageous properties, such as low mechanical resistance and susceptibility to harmful water action. This is a serious problem especially for masonry mortars, whereas adherence behavior, shrinkage and color should be considered in designing rendering mortars. Also salt crystallization, hygroscopicity and hydration are among disruptive factors negatively affecting the durability of lime-based mortars [20,21]. Besides that, building practice impose demands on a complex hygrothermal performance of mortars for repair purposes what take into consideration application of lightweight materials that can partially improve thermal behavior of renovated buildings.

In this context, blended mortars represent alternative as they are able to combine the advantages of both aerial lime and hydraulic binders. Based on literature review, the most-studied mortars were those based on lime-cement blends [22,23,24,25]. However, use of Portland cement in renewal of historical masonry is strictly forbidden by culture heritage authorities due to its serious incompatibility with original materials that caused in recent restoration inventions significant failures and accelerated the damage of the restored monuments [11].

Instead of Portland cement, materials rich in amorphous silica and/or alumina find use as components of lime blends. It is in accordance with composition of mortars produced by early civilizations that used ceramic and volcanic dust to improve the properties and durability of lime mortars. These materials are called pozzolans, and when appropriately milled react in the presence of water with Ca(OH)_2_ and form hydraulic products having improved durability characteristics [26]. Pozzolans are of the natural origin (diatomite, zeolite, pumice, etc.) or are produced artificially, for example by thermal processing (metakaolin, perlite). Pozzolans are also by-products of coal and agricultural products combustion (coal fly ash, rice husk ash, pam ash, sugar cane ash, etc.). The cement-pozzolan mortars were intensively studied during last 30 years [27,28,29]. On the other hand, studies on the use of pozzolans in lime-based masonry or rendering mortars are relatively rare. Sala et al. [30] analyzed properties of lime mortar with natural pozzolan coming from the complex of Monti Sabatini (Rome, Italy). The reported tests clearly showed increase in the mortar strength with the addition of pozzolana. Pavia and Aly [31] studied hydrated lime mortars with rice husk ash and ground granulated blast furnace slag. Both mineral admixtures increased mortars strength with little impact on hygric properties. Ulukaya and Yuzer [32] investigated crushed brick-lime mortars. They observed pozzolanic activity of crushed bricks that resulted in a significant improvement of mortar mechanical properties. Stefanidou [33] used two natural pozzolans in combination with lime for production of repair mortars. Based on the above given survey it can be summarized, the use of lime and pozzolans blends in composition of repair mortars is indispensable for achieving their sufficient mechanical resistance and durability.

In recent decades, there has been worldwide increased awareness of the used of pozzolanic materials in production of materials for construction industry. Nevertheless, specific attention must be paid to renewable mineral admixtures and materials that were formerly considered as waste. As building industry is big consumers of natural resources that are used especially for concrete, cement, and lime production, their partial substitution by any suitable eco-efficient material is highly positively appreciated. Therefore, the biomass ash coming from wood chips combustion was studied in this paper as an active eco-efficient mineral admixture in repair mortar based on lime hydrate as the main binder. The potential of the use of biomass ash coming from combustion of different agricultural products as pozzolanic material was proved recently by several research studies conducted on the use of biomass ash as partial PC substitution in mortar and concrete mix design [34,35,36]. In [34] the authors applied biomass ash coming from wood chips ash production in composition of cement mortar. The strength activity index was high for all examined materials what proved the wood chips ash pozzolanity. However, based on our previous research conducted on biomass ashes, one must always consider origin, chemical and physical parameters of the particular biomass ash before its use as pozzolan.

The other pozzolan used in ternary blended binder was metakaolin, because of its high pozzolanity and proved benefits of its use in lime mortars [37,38]. The pozzolanic activity of metakolin evaluated, e.g., Bakolas et al. [39]. Based on pozzolanic activity tests and assessed physicomechanical characteristics of lime-metakaolin pastes, the compounds formed within the pozzolanic reaction were CSH and C_2_ASH_8_. Moreover, the calcium hydroxide consumption was higher as the initial dosage of metakaolin in paste increased. Aggelakopoulou et al. [40] studied properties of lime-metakaolin mortar for the restoration of historical masonry. Authors reported on the increased static modulus of elasticity, compressive and flexural strength, with the increased metakaolin/lime ratio in mortar mix.

The novelty of the paper lies in a unique combination of biomass ash and metakaolin in composition of ternary blended binder for production of repair mortars. Based on literature survey given above, both these materials proved their pozzolanic character and applicability in cement-based composites. Moreover, metakaolin found its use also in lime mortars for repair and renovation of historical buildings. In this respect, their combined use represents innovative solution for design and development of a novel type of repair mortar that maintained excellent properties of lime mortars and possessed improved mechanical resistance and durability. Besides that, the reuse of biomass ash in production of construction materials brings both economic and environmental benefits. In this respect, the novel repair mortar can be considered as a cost effective material that can be produced in more sustainable way than most of mortars already used in building practice.

## 2. Materials and Methods

### 2.1. Materials and Samples Preparation

The biomass ash (BA) was obtained from Plzeň heating plant, Plzeň, Czech Republic. The BA examined in this work was a product of wood chips combustion and was used in mortar mix as a part of blended binder. Another binder component was commercially produced metakaolin Mefisto K05 (ČLUZ a.s., Nové Strašecí, Czech Republic). As a primary binder, lime hydrate coming from lime kiln Čertovy schody a.s., Tmaň, Czech Republic, was used. As aggregate, silica sand of fraction 0–2 mm was used. It is a product of Filtrační písky, spol. s r.o., Chlum u Doks, Czech Republic and it was mixed from three fractions. The weight ratio of sand fractions was 1:1:1. The reference lime mortar (M_ref_) was tested as well. The composition of studied materials is summarized in Table 1.

In both mortar mixes, the binder/aggregate ratio was 1/3. The water/binder ratio was high as we planned to developed mortars of high porosity what can be simply achieved by a higher amount of batch water. The water/binder ratio was adjusted to get a similar consistency of both mortars. It was verified by flow table test. The value of spreading was 160/160 mm. As BA increased mortar workability, the water/binder ratio was decreased in MBAM mortar.

The casted specimens were rectangular prisms having dimensions of 40 mm × 40 mm × 160 mm, cubes of side 100 mm, and circular flat specimens with diameter of 100 mm and thickness of 20 mm. Before the particular tests, they were stored in a humidity of approx. 98% and temperature (23 ± 2) °C for 28 days or 365 days respectively. During curing, the specimens were placed on a steel grid in order to allow their carbonation.

### 2.2. Characterization of BA and Metakaolin

For BA and metakaolin, specific density and Blaine specific surface were measured. The specific density was tested using a Pycnomatic ATC (Thermo Scientific, Milan, Italy). Blaine specific surface was assessed by a Blaine device [41].

For the determination of pozzolanic activity of BA and metakaolin, Chapelle test was used [42].

Chemical composition of BA and Mefisto K05 was measured by X-ray fluorescence (XRF) using Axios sequential WD-XRF spectrometer (PANalytical, Almelo, The Netherlands).

Morphology of binder constituents was studied by scanning electron microscope (TESCAN Lyra 3, Tescan Brno, s.r.o., Brno, Czech Republic) equipped with a FEG electron source. Energy dispersive spectroscopy (EDS) was performed with the analyzer X-MaxN, SDD detector (Oxford instruments, High Wycombe, UK) and AZtecEnergy software (Oxford instruments) to determine the chemical composition.

X-ray diffraction (XRD) was performed on Bruker D8 Discoverer (Bruker AXS GmbH, Karlsruhe, Germany) powder diffractometer with parafocusing Bragg–Brentano geometry at room temperature using CuK_α_ radiation [34].

The particle size distribution was tested using an apparatus Analysette 22 Micro Tec plus (Fritsch, Idar-Oberstain, Germany) working on a laser diffraction. The device allows identification of particles having size from 0.08 μm to 2 mm.

### 2.3. Methods of Testing Hardened Mortar Samples

The tests conducted were aimed at the assessment of structural parameters, mechanical resistance, hygrothermal performance and simultaneous thermal analysis (STA). In addition to the tests conducted for 28 days samples, measurement of basic structural and mechanical parameters was performed also for 365 days samples. The overview of the experimental program reporting type of test, maturation time and number of specimens is given in Table 2.

Among fundamental structural properties of researched mortars, bulk density, specific density, and total open porosity were tested. Before these tests, the specimens were dried in a vacuum drier at 60 °C. The specific density *ρ_s_* (kg/m^3^) was measured using helium pycnometry. The bulk density *ρ_b_* (kg/m^3^) of mortar specimens was determined according to the EN 1015-10 [43] using gravimetric method. The expanded combined uncertainty of the bulk density test was 1.2%. The total open porosity *ψ* (%) was calculated from the known bulk density and matrix density values with combined uncertainty of 1.7% [34].

As also pore size affects material performance from the mechanical resistance and hygrothermal performance point of view, pore size distribution was measured using mercury porosimetry (Pascal 140 and Pascal 440, Thermo, Milan, Italy). The typical specimens mass was about 0.9 g.

For the assessment of mechanical resistance of mortars with admixing of BA and metakaolin, their flexural strength *f_f_* (MPa), compressive strength *f_c_* (MPa), and dynamic Young’s modulus *E_d_* (GPa) were tested. The strength tests were conducted according to the standard EN 1015-11 [44] on 40 mm × 40 mm × 160 mm prisms and their rests from flexural strength testing. The relative expanded uncertainty of strength tests was 1.4%. The Young’s modulus *E_d_* (GPa) was tested using a DIO 562 apparatus (Starmans Electronics s.r.o., Prague, Czech Republic). The expanded combined uncertainty of this test method was 5.6%. Based on the compressive strength data, strength activity index (SAI) was calculated [45]. It is generally accepted, a material can be considered as pozzolan if SAI ≥ 75% [46].

As masonry and rendering mortars usually suffer from excessive moisture presence, ability of studied mortars to transport water and water vapor was tested. Water absorption coefficient *A_w_* (kg/(m^2^s^1/2^)) and saturated moisture content *w_sat_* (kg/m^3^) were measured in free water intake experiment [47] that was conducted according to the standard EN 1015-18 [48]. Apparent moisture diffusivity *κ* (m^2^/s) was calculated as originally proposed by Kumaran [49]. The expanded combined uncertainty of the water absorption test was 2.3%, of the saturated moisture content test it was 1.4% and that of the apparent moisture diffusivity test was 6.6%.

Water vapor transport parameters were tested under isothermal conditions using cup method. The water vapor transmission test was performed following standard ISO 12572 [50] in the both dry-cup and wet-cup arrangements. Details on the measurement conditions can be found in [34]. Based on the cup test, the water vapor permeability *δ* (s), the water vapor diffusion coefficient *D* (m^2^/s), and the water vapor resistance factor *μ* (-) were assessed. The expanded combined uncertainty of the cup test was for 1.4% for *δ*, 1.6% for *D*, and 2.1 for *μ*.

Because renders have a significant effect on the thermal performance of buildings, thermal transport and storage parameters of studied mortars were tested. For the dry and fully water saturated thermal conductivity *λ* (W/(m·K)) and volumetric heat capacity *c_v_* (J/(m^3^·K)) measurement, ISOMET 2114 (Applied Precision) was used. The expanded combined uncertainty of the thermal conductivity and volumetric heat capacity tests was 1.7%.

Thermal behavior of tested mortars was analyzed by Simultaneous thermal analysis (STA) on a Linseis STA PT1600 apparatus (Linseis Messgeraete GmbH, Selb, Germany).

Mortar specimens were also studied using Dino-Lite digital optical microscope (Dino-Lite/IDCP B.V., Naarden, The Netherlands) with resolution of 5 Mpx.

## 3. Results and Discussion

Basic structural parameters and pozzolanic activity of binder constituents is given in Table 3. For comparison, Blaine fineness and specific density of lime hydrate were also introduced. The lowest Blaine fineness exhibited biomass ash. It was due to its insufficient milling in laboratory contrary to effective milling of commercially manufactured lime hydrate and metakaolin. On the other hand, both mineral admixtures proved their high pozzolanic activity considering the lower limit of 650 mg of Ca(OH)_2_/g of material to be classified as pozzolana active [51,52]. 

Chemical composition of BA and metakaolin obtained by XRF analysis is given in Table 4, whereas normalized wt.% values are presented. Biomass ash and metakaolin contained high amount of silica and alumina phases what was positive for their presumed use as active mineral admixtures in mortar mix composition. The content of chlorides and sulfates was in both studied materials low, i.e., new resources of soluble salts rising from the use of repair mortar with incorporated BA and Mefisto K05 are avoided, being salt action one of the most often causes of decay of old masonry [53].

The morphology and elemental distribution maps of BA and metakaolin are apparent from SEM images and EDS mapping in Figure 1. SEM micrographs of BA proved varied morphology; particles had dimensions between 1 μm and 50 μm. These particles often form huge agglomerates. Also their shapes differed. Obtained chemical composition was in good agreement with XRF results: the most represented elements were C, O, Si, Ca, K, Al, Mg, Fe, and Na. On the other hand, SEM micrographs of Metakaolin showed less agglomerates. Particles had dimensions between 1 μm and 10 μm. EDS confirmed that the major elements present were O, C, Si, Al, and K in this sample.

Phase composition of BA and metakaolin was analyzed using X-Ray diffraction (XRD) (see Figure 2).

Sample BA was highly crystalline and it contained mainly silicon dioxide (quartz), calcium carbonate (calcite), and also other aluminosilicates. Metakaolin was highly amorphous, but some crystalline phases were detected as well (quartz, mullite, and others).

Particle size distribution of lime hydrate, BA, and metakaolin is graphed in Figure 3 and Figure 4. Metakaolin Mefisto K05 was the finest binder constituent (*d*_50_ = 3.5 μm) what was in the agreement with its high specific surface. BA milled in laboratory had slightly worse fineness (*d*_50_ = 34.3 μm) compared to lime hydrate (*d*_50_ = 31.2 μm). However, it was still acceptable considering the upper limit for pozzolan fineness introduced in EN 450-1 [45].

Photography of prepared samples is shown in Figure 5A. This figure demonstrated that both samples contained minimum cracks or other defects on both macroscales and microscales. In comparison, sample MBAM has even lower amount of defects in comparison to the reference M_ref_. Figure 5B shows micrographs obtained by optical microscopy. In both cases, phases are homogenously distributed, no agglomerates were detected. The color of mortar with incorporated biomass ash was slightly changed compared to the white-beige color of reference mortar. It was bright gray due to the use of biomass ash. For masonry mortar, the color change is not a crucial parameter. On the other hand, for rendering purposes, the gray color of the developed mortar can be beneficially used for sgraffito technique for a facade decoration. Moreover, the color of biomass ash and thus of mortar with ternary based binder can be changed by its thermal treatment.

The presented material parameters of hardened mortars represent average value obtained from the measurement of a specific number of samples studied in the particular research test (see Table 2). The fundamental structural properties of researched mortars are summarized in Table 5. The use of BA and metakaolin resulted in decrease in bulk density and thus significant increase in porosity despite of a lower water/binder ratio of MBAM mortar compared to reference mix. One of the decisive parameters of repair mortars is their high porosity that should enable possible salt accumulation and water vapor transmission. According to the WTA-Specifications (International Association for the Science and Technology in Maintenance of Structures and Protection of Monuments) the porosity of repair mortar must be >40 % [54]. In this respect, the use of ternary blended binder was quite effective and the obtained data proved its applicability in design of rendering and masonry mortars applicable for historical damp and salt contaminated interior or exterior walls.

Pore size distribution measured by mercury intrusion porosimetry is shown in Figure 6.

The total porosity assessed by mercury intrusion porosimetry was 43.1% for MBAM and 32.8% for M_ref_. Considering the mass of tested specimens which was about 0.9 g, the results were in quantitative agreement with those calculated on the basis of specific density and bulk density tests. MBAM had significantly higher volume of pores in the range from 0.01 μm to 1.3 μm. According to IUPAC (International Union of Pure and Applied Chemistry) classification of pore sizes [55], these pore were classified as macropores (large capillaries > 0.05 μm) and mesopores (medium capillaries 0.05–0.01 μm). Macropores affected mortar strength and mesopores its transfer properties. On the other hand, reference mortar M_ref_ exhibited higher volume of macropores >1.3 μm, what was probably the main reason of its lower strength. The volume of small gel capillaries (<0.01 μm) was in both studied mortars negligible as these pores are common for hydraulic mortars [56]. In lime mortars are gel pores limited and are formed inside the binder crystal lattice [57].

The mechanical parameters of tested mortars are given in Table 6. In spite of higher porosity of mortar MBAM, its mechanical resistance was higher compared to that of reference mortar M_ref_. This feature was apparent for both 28 days and 365 days samples. In this case, the porosity was not only parameter contributing to the mechanical resistance of the developed ternary binder-based mortar. The increase in mechanical resistance we assign to C-S-H and C-S-A-H phases formed within the pozzolanic reaction of metakaolin and biomass ash with lime hydrate as these products are more durable and mechanical resistant compared to products of lime hydrate hardening. Quantitatively, values of mechanical parameters obtained for 28 days samples were higher than considered typical for non-hydraulic mortars [58,59] what is promising for mortar use not only for rendering but also for structural (walling) purposes. The value of strength activity index was high, and gave information on great pozzolanic activity of both incorporated mineral admixtures.

The parameters characterizing liquid water transport in tested mortars are presented in Table 7. Their values are in agreement with porosity data, i.e., due to the increased total open porosity the water absorption accelerated. This feature is on the one hand positive, as it ensures water transport to areas when evaporation takes place, and on the other hand, it can negatively affect mortars and their substrates durability in the presence of excessive moisture. Therefore, in practical applications, the exterior surface of renderings mortars made of MBAM mortar type must be provided with hydrophobic additive or the inner hydrophobization of MBAM must be applied.

The water vapor transition parameters are given in Table 8. As reported by other researchers [60] water vapor transport in case of wet-cup arrangement of the cup test was faster compared to dry-cup test. This can be attributed to filling of pore surface with water molecules what reduced binding forces decelerating water vapor transmission. The MBAM mortar exhibited significantly lower water vapor resistance factor compared to reference material. This was due to its high open porosity. Slightly lower values of water vapor permeability reported, e.g., Barnat-Hunek et al. [61] who tested properties of cement-lime restoration renders produced with zeolite, lightweight aggregate, and boiler slag. This mortar performance is highly promising for its application on damp and salt laden masonry as high water vapor permeability and water vapor diffusion coefficient ensure water vapor evaporation, and thus natural drying of such exposed structures. Moreover, the water vapor resistance factor of MBAM was safely < 15, what is limiting value required by EN 988-1 [62] for repair mortars. In this respect, the use of ternary based binder whose design and testing for application in repair mortars is the main goal of the paper, was quite successful for achievement high water vapor transmission rate of the developed mortar.

Mortars thermal properties are presented in Table 9. One must take into consideration, mortar as part of building envelopes contributes significantly to its overall hygrothermal performance. Therefore, its low thermal conductivity and moderate heat capacity are often required in repair works. As expected, high porosity of mortar MBAM reduced its thermal conductivity as well as volumetric heat capacity. The dry thermal conductivity value of MBAM was even more than two times lower compared to that obtained for reference mortar M_ref_. This mortar behavior can find use in design of thermal insulation renders or repair mortars with improved thermal performance. The presence of water increased the both investigated thermal parameters as water has high thermal conductivity and volumetric heat capacity compared to dry air [63]. On similar increase in thermal conductivity with increasing moisture content reported, e.g., Pavlík et al. [64] who tested thermal properties of lime-pozzolan plasters based on waste ceramic powder.

Thermal behavior of prepared samples M_ref_ and MBAM was tested using STA (see Figure 7). Two major effects were detected for the sample M_ref_. The first endothermic effect starting at ~400 °C is the decomposition of portlandite, during the decomposition water is released as can be seen on the TG curve (weight decrease 3.2%). The onset temperature of the second endothermic effect was at ~700 °C. This effect was caused because of the decomposition of calcium carbonate. The release of carbon dioxide was clearly visible from the TG curve (weight decrease 13.8%). Thermal behavior of the sample MBAM was different: the first exothermic effect was caused due to oxidation of remaining carbon that was resent in biomass ash. This effect was accompanied by a gradual decrease in weight. The second effect, similarly to M_ref_, was caused by the decomposition of calcium carbonate. In this case it was difficult to exactly determine the weight decrease due to the shape of TG curve, decrease was approx. 8.0%. The overall weight decrease of both samples during the heating was comparable.

Repair mortars for use in historical building must fulfil compatibility criteria with respect to original materials and qualitative criteria required by technical standards. The conducted experimental campaign aimed at the assessment of a novel type of lime-pozzolan ternary blended binder applied in mortar composition proved that studied material met all considered qualitative criteria for repair mortars. Nevertheless, a satisfactory evaluation of the suitability of the mortars for restoration is quite difficult and specific parameters of materials for repair purposes must be considered case by case taking into account properties of original inbuilt materials, their state, range of damage, exterior and interior service conditions.

## 4. Conclusions

In this study, new type of ternary blended binder was studied as possible binding material for production of a novel repair mortar. The ternary binder was mixed from lime hydrate, biomass ash, and metakaolin. On the basis of obtained experimental results, the following conclusions can be drawn:Biomass ash and metakaolin contained high content of silica and alumina phases what was positive for their presumed use as active mineral admixtures in mortar mix composition.Both alternative binder components exhibited high pozzolanic character and sufficient fineness for their blending with lime hydrate.Mortar with ternary blended binder showed decreased bulk density and thus significantly increased porosity. As one of the decisive parameters of repair mortar is their high porosity (>40%), the use of ternary blended binder was quite effective in this respect.The value of strength activity index was for mortar MBAM high, and gave evidence of great pozzolanic activity of both incorporated mineral admixtures.The water vapor resistance factor of mortar MBAM met the criteria required of EU standards for repair mortars.The use of ternary blended binder resulted in a significant decrease in the thermal conductivity. Based on that, the developed mortar MBAM can contribute to the improvement of the overall hygrothermal performance of renovated masonry.

As the developed lightweight mortar exhibited good functional performance it can be concluded, the tested biomass ash could potentially represent an eco-efficient low-cost alternative to other pozzolans commonly used in construction industry. Moreover, biomass ash blending with metakaolin and lime hydrate gave binder acceptable by culture heritage authorities for restoration of historical buildings and monuments. Because construction industry suffers from the limited natural resources used for production of building materials, reuse of biomass ash as by product of energy generation in composition of ternary blended binder represents beneficial solution for ash disposal considering circular economy principles. For the practical use of mortar MBAM examined in this paper, it will be necessary to apply water repellents, workability, adhesive, and other additives to adjust its properties for specific rendering and structural purposes. Specific attention must be also paid to the formulation of mortar mix, considering BA and metakaolin dosage, e.g., ratio of lime hydrate replacement, and water/binder ratio. As building practice impose in repair historical buildings complex requirements on material hygrothermal and mechanical performance, the problem of a possible use of lightweight aggregate (perlite, zeolite, pumice, expanded glass granulate, etc.) in mortar composition should be also addressed. In this way, it will be possible to partially improve thermal behavior of repaired buildings without application of any additional thermal insulation systems that are mostly forbidden in historical listed buildings. These analyses, experiments and tests will be the subject of the future applied research that will be necessary before launch the newly developed repair mortar in the market.

## Figures and Tables

**Figure 1 materials-12-00996-f001:**
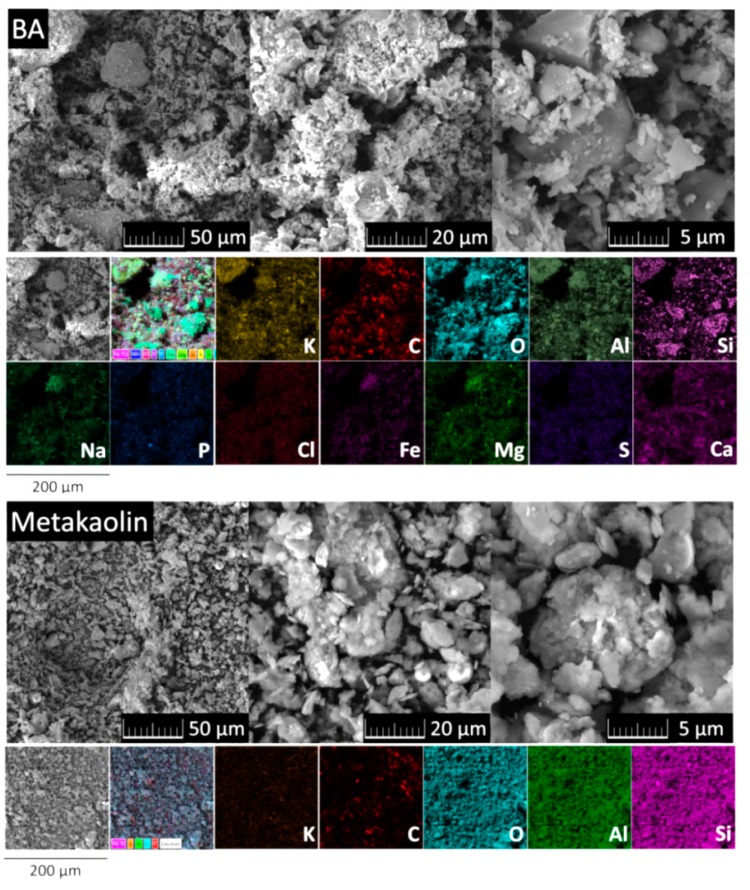
Morphology and elemental distribution maps of BA and metakaolin obtained by scanning electron microscopy energy dispersive spectroscopy (SEM-EDS).

**Figure 2 materials-12-00996-f002:**
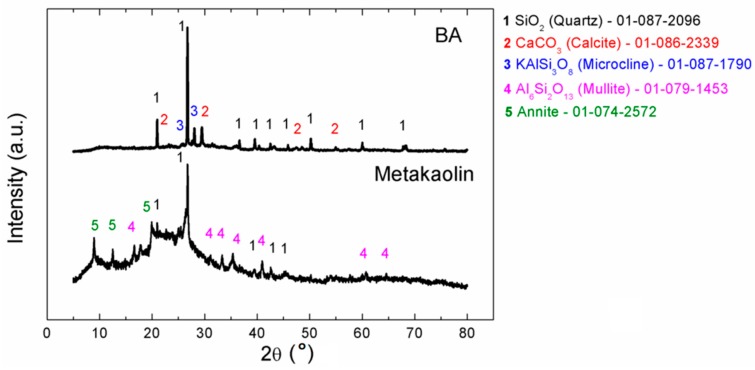
Diffractograms of BA and Metakaolin.

**Figure 3 materials-12-00996-f003:**
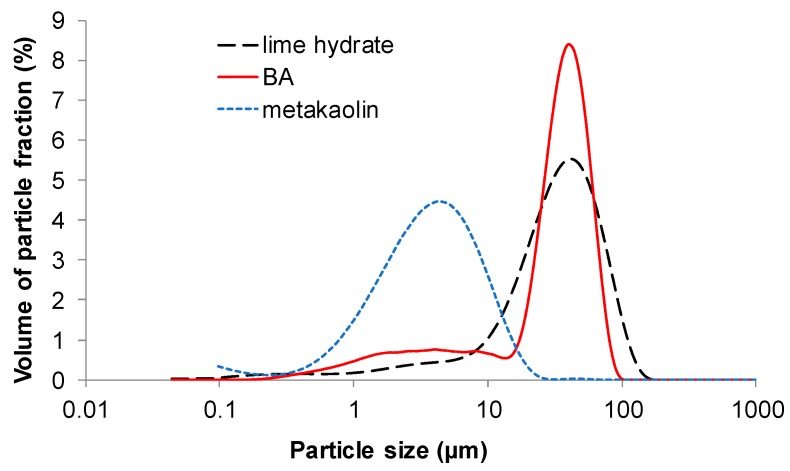
Particle size distribution—distribution curves.

**Figure 4 materials-12-00996-f004:**
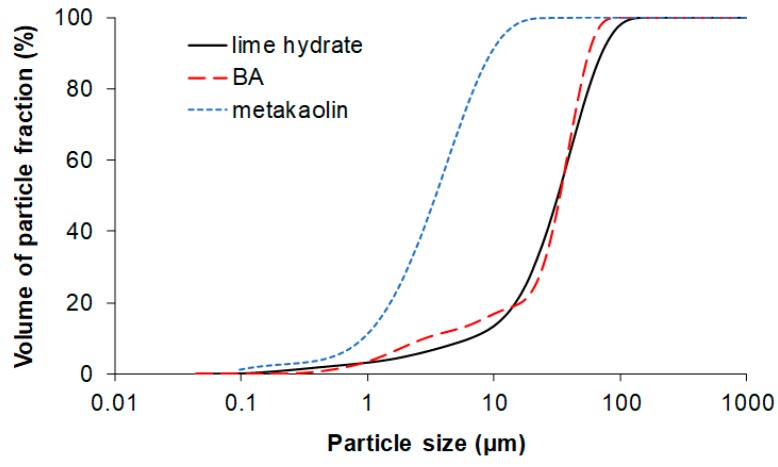
Particle size distribution—cumulative curves.

**Figure 5 materials-12-00996-f005:**
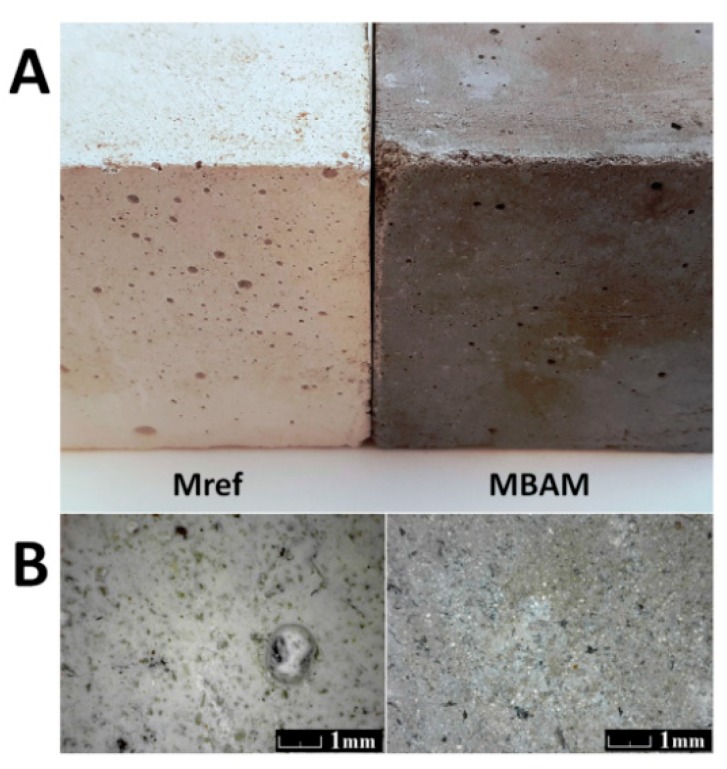
(**A**) Photography of M_ref_ (**left**) and MBAM (**right**) and (**B**) optical micrograph of M_ref_ (**left**) and MBMA (**right**).

**Figure 6 materials-12-00996-f006:**
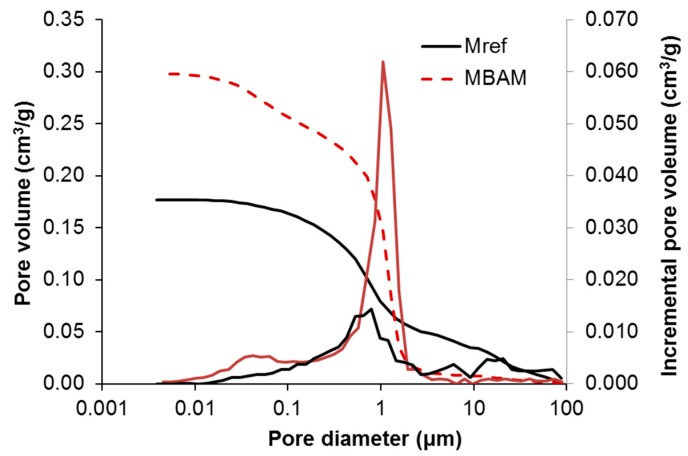
Pore size distribution.

**Figure 7 materials-12-00996-f007:**
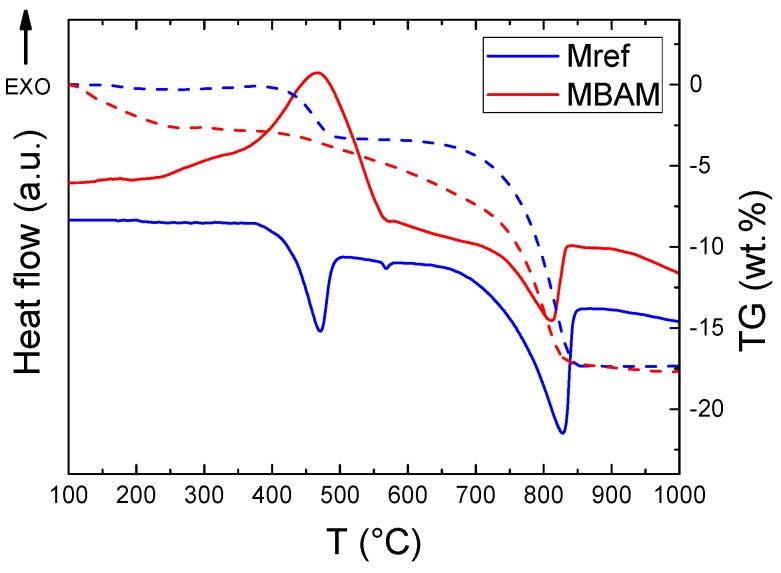
DTA a TG curves of (**A**) M_ref_ and (**B**) MBAM measured in air atmosphere.

**Table 1 materials-12-00996-t001:** Composition of studied mortars, mass (g). BA: biomass ash.

Mortar Mix	Lime Hydrate	BA	Metakaolin	Silica Sand	Water	w/b (-)
M_ref_	1350	-	-	3 × 1350	1350	1.0
MBAM	643	578	129	3 × 1350	1202	0.9

MBAM stands for mortar with ternary based binder based of lime hydrate, biomass ash and metakaolin.

**Table 2 materials-12-00996-t002:** Overview of the experimental campaign conducted for hardened mortar samples.

Test	Maturation Time (days)	Number of Specimens
Bulk density	28, 365	5
Specific density	28, 365	5
Open porosity	28, 365	5
Pore size distribution	28	2
Flexural strength	28, 365	3
Compressive strength	28, 365	6
Young’s modulus	28, 365	5
Water absorption coefficient	28	5
Saturation moisture content	28	5
Apparent moisture diffusivity	28	5
Water vapor permeability	28	5
Water vapor diffusion coefficient	28	5
Water vapor resistance factor	28	5
Volumetric heat capacity	28	5
Thermal conductivity	28	5
simultaneous thermal analysis (STA)	28	2

**Table 3 materials-12-00996-t003:** Physical parameters and pozzolanic activity of binder constituents.

Materials	Blaine Fineness (m^2^/kg)	Specific Density (kg/m^3^)	Pozzolanic Activity (%)
Lime hydrate	2205	2214	-
BA	798	2614	1340
Mefisto K05	1608	2622	1967

**Table 4 materials-12-00996-t004:** X-Ray fluorescence analysis (XRF) chemical composition, amount (wt.%).

Substance	BA	Mefisto K05
CuO	0.01	-
Na_2_O	0.31	-
ZnO	0.11	-
MgO	1.18	0.44
Al_2_O_3_	12.32	37.91
BaO	0.16	-
SiO_2_	48.72	59.41
P_2_O_5_	2.63	-
SO_3_	1.24	-
SrO	0.07	0.03
Cl	0.24	-
ZrO_2_	0.04	-
K_2_O	7.56	0.81
CaO	17.45	0.20
TiO_2_	0.82	0.51
V_2_O_5_	0.03	-
PbO	0.03	-
MnO	1.66	-
Fe_2_O_3_	5.42	0.72
∑	100.00	100.00

**Table 5 materials-12-00996-t005:** Fundamental structural properties.

Materials		Specific Density *ρ*_s_ (kg/m^3^)	Bulk Density *ρ**_b_* (kg/m^3^)	Total Open Porosity *ψ* (%)
28 days	M_ref_	2560	1670 ± 20	34.7 ± 0.6
	MBAM	2500	1410 ± 17	43.6 ± 0.7
365 days	M_ref_	2580	1710 ± 21	33.7 ± 0.6
	MBAM	2540	1475 ± 18	41.9 ± 0.7

**Table 6 materials-12-00996-t006:** Mechanical properties.

Materials		Compressive Strength *f_c_* (MPa)	Flexural Strength *f_f_* (MPa)	Young’s Modulus *E_d_* (GPa)	SAI (%)
28 days	M_ref_	1.32 ± 0.02	0.47 ± 0.007	2.6 ± 0.15	-
	MBAM	1.56 ± 0.02	0.51 ± 0.007	2.7 ± 0.15	118.2
365 days	M_ref_	1.78 ± 0.03	0.72 ± 0.01	2.9 ± 0.16	-
	MBAM	1.98 ± 0.02	0.75 ± 0.01	3.0 ± 0.17	111.2

**Table 7 materials-12-00996-t007:** Water transport properties.

Materials	*A_w_* (kg/(m^2^·s^1/2^))	*w_sat_* (kg/m^3^)	*κ_app_* × 10^−7^(m^2^/s)
M_ref_	0.324 ± 0.008	341 ± 5	9.0 ± 0.6
MBAM	0.423 ± 0.010	433 ± 6	9.5 ± 0.6

**Table 8 materials-12-00996-t008:** Water vapor transmission properties.

Materials		*δ* × 10^−11^ (s)	*D* × 10^−6^ (m^2^/s)	*μ* (-)
dry-cup	M_ref_	1.50 ± 0.02	2.05 ± 0.03	12.2 ± 0.3
	MBAM	1.78 ± 0.03	2.43± 0.04	10.3 ± 0.2
wet-cup	M_ref_	1.66 ± 0.02	2.27 ± 0.04	11.0 ± 0.2
	MBAM	2.10 ± 0.03	2.87± 0.05	8.7 ± 0.2

**Table 9 materials-12-00996-t009:** Thermal properties.

Materials	*λ_dry_* (W/(m·K))	*c_vdry_* × 10^6^ (J/(m^3^·K))	*λ_sat_* (W/(m·K))	*c_vsat_* × 10^6^ (J/(m^3^·K))
M_ref_	1.10 ± 0.02	1.60 ± 0.03	3.29 ± 0.06	1.89 ± 0.03
MBAM	0.45 ± 0.008	1.48 ± 0.03	2.18 ± 0.04	2.09 ± 0.04

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
