# Peer review of "Ternary Blended Binder for Production of a Novel Type of Lightweight Repair Mortar"

_materials, 2019, doi:10.3390/ma12060996_

Reviewer 1 Report

This manuscript focuses on the obtaining of a new repair mortar based on air lime modified with metakaolin and biomass ash as pozzolanic additives. Generally speaking, the paper is well written and well structured, and from the scientific and technical point of view it is a novel and interesting contribution. The idea of testing and developing new lime-based repair materials is of interest for preserving the Architectural Heritage, so that any research on this topic enlarging the knowledge on this area is welcome.

However, the paper needs a revision on several points that should be better explained or discussed.

1.       The authors should revise what kind of mortars are referring to. Sometimes it appears that the manuscript focuses on rendering mortars and plasters. It is OK, but, in this case, mechanical properties are not particularly significant. Instead, adherence behavior, shrinkage and color should have been considered.

In this line, revise the sentence in line 58 about the mechanical resistance requirements for rendering mortars and plasters and add a discussion about the properties aforementioned. It seems to me that the color of some of the obtained specimens could be a serious drawback for using these mortars in repair works of the Built Heritage. Please, add some comments on this point.

2.       The term “renovation” is acceptable, although “repair” mortar is much more common. Please, consider the appropriateness of the suggestion.

3.       The literature review about the use of pozzolans in lime-based mortars should be improved. You will find previous works dealing with metakaolin and air lime mortars and nanosilica and air lime mortars, for example, that should be considered in the state-of-the-art.

4.       At the end of the Introduction, you should add the rationale behind the use of this biomass ash from wood chips combustion. Has been this BA previously used and tested as pozzolanic material? What is your previous knowledge on its composition supporting its use as pozzolanic material? Add comments on this Biomass Ash, its composition, its expected pozzolanic performance and its effects according to previous works in scientific literature.

5.       In section 3.1 you explain that BA was used as a part of the blended binder. In the Abstract, you explain that BA was used as partial substitute of the hydrated lime. In Table 1 you should revise and specify exactly the percentages of substitution of hydrated lime by both BA and Metakaolin. Current amounts are hardly understandable. The binder/aggregate amounts do not match the values reported for hydrated lime, BA, metakaolin and silica sand.

6.       Did you realize that the selected curing conditions are good for hydraulic mortars but are not suitable for your reference mortar (air lime mortar)? Add some comments on this and relate them with the mechanical performance discussed in lines 279-286: the BA mortars can reach a mechanical resistance higher than that of the reference mortar because the inadequate curing conditions for air lime mortar (the reference one).

7.       Please, revise the mistake in the whole text and write Blaine instead of Blain.

8.       Section 3.2, add XRD measurement conditions (angles, step size, …).

9.        Line 267, delete the number 6 in Figure 5,6.

10.   Line 223, revise the mistake and delete the final “e” in metakaoline.

11.   Line 342: please, write better “high pozzolanic character”.

Author Response

We highly appreciate work of reviewer 1 on review of our manuscript. We addressed all the reviewer’s comments and followed reviewer’s suggestions in the revised version of the manuscript. We believe that reviewer’s comments and suggestions have contributed substantially to the improvement of the presentation of our study and the overall quality of the manuscript. The revised text is in a revised manuscript marked up by a red colour.

Reviewer 1:

This manuscript focuses on the obtaining of a new repair mortar based on air lime modified with metakaolin and biomass ash as pozzolanic additives. Generally speaking, the paper is well written and well structured, and from the scientific and technical point of view it is a novel and interesting contribution. The idea of testing and developing new lime-based repair materials is of interest for preserving the Architectural Heritage, so that any research on this topic enlarging the knowledge on this area is welcome.

However, the paper needs a revision on several points that should be better explained or discussed.

We thank reviewer 1 for considering our paper well written and structured. We highly appreciate that reviewer 1 evaluate our paper as novel and interesting from the scientific point of view.

Point 1: The authors should revise what kind of mortars are referring to. Sometimes it appears that the manuscript focuses on rendering mortars and plasters. It is OK, but, in this case, mechanical properties are not particularly significant. Instead, adherence behaviour, shrinkage and colour should have been considered.

In this line, revise the sentence in line 58 about the mechanical resistance requirements for rendering mortars and plasters and add a discussion about the properties aforementioned. It seems to me that the colour of some of the obtained specimens could be a serious drawback for using these mortars in repair works of the Built Heritage. Please, add some comments on this point.

We agree with the reviewer 1, the explanation what kinds of mortars were studied was not clear. We studied mortars that can be used both as structural mortars for renovation of historical masonry and also as rendering materials. It is right that mechanical properties are not of the particular importance for rendering materials, but their improvement by the use of pozzolanic materials is always considered good in general especially from the point of view of their durability.  Adherence can be in final product improved by commercial additives, what was not subject of this let us say “basic research”.

The colour of mortar with incorporated biomass ash was slightly changed compared to white-beige colour of reference mortar. The mortar with ternary blended binder was bright grey due to the use of biomass ash. For masonry mortar, the colour change is not crucial parameter. On the other hand, for rendering purposes, the grey colour of the developed mortar can be beneficially used for sgraffito technique for a facade decoration. Moreover, the colour of biomass ash and thus of mortar with ternary based binder can be changed by its thermal treatment.        

Based on reviewer’s comments we corrected manuscript accordingly.

Point 2: The term “renovation” is acceptable, although “repair” mortar is much more common. Please, consider the appropriateness of the suggestion

We agree with reviewer 1, the term repair is more common. As suggested, we replaced term “renovation” by “repair” throughout the manuscript as well as in paper title.

Point 3: The literature review about the use of pozzolans in lime-based mortars should be improved. You will find previous works dealing with metakaolin and air lime mortars and nanosilica and air lime mortars, for example, that should be considered in the state-of-the-art.

As required, we completed state-of-the-art section with literature review about metakolin use in lime-based mortars. In this respect, we also added new references [37-40].

Point 4: At the end of the Introduction, you should add the rationale behind the use of this biomass ash from wood chips combustion. Has been this BA previously used and tested as pozzolanic material? What is your previous knowledge on its composition supporting its use as pozzolanic material? Add comments on this Biomass Ash, its composition, its expected pozzolanic performance and its effects according to previous works in scientific literature.

We improved this paper section as proposed by reviewer 1. Based on out previous research when we studied different types of biomass ashes we proved the high pozzolanic activity of wheat straw and wood chips ashes. We also newly added reference [34-36].

Point 5: In section 3.1 you explain that BA was used as a part of the blended binder. In the Abstract, you explain that BA was used as partial substitute of the hydrated lime. In Table 1 you should revise and specify exactly the percentages of substitution of hydrated lime by both BA and Metakaolin. Current amounts are hardly understandable. The binder/aggregate amounts do not match the values reported for hydrated lime, BA, metakaolin and silica sand.

Yes, BA was used as a part of the blended binder, e.g., partial substitute of the hydrated lime. Nevertheless, data introduced in Table 1 were not converted to the volume of fresh mortar mix. Therefore, data in Table 1 were changed in order to present it in a clear way. The idea of the mix design was to maintain similar binder/aggregate ratio in both reference and modified mortars mixes. The water/binder ratio was adjusted to get similar workability of both mortars. It was verified by flow table tests. The value of spreading was 160/160 mm. As BA increased mortar workability, the water/binder ratio was decreased in MBAM mortar.

Point 6: Did you realize that the selected curing conditions are good for hydraulic mortars but are not suitable for your reference mortar (air lime mortar)? Add some comments on this and relate them with the mechanical performance discussed in lines 279-286: the BA mortars can reach a mechanical resistance higher than that of the reference mortar because the inadequate curing conditions for air lime mortar (the reference one).

Based on out previous research we are using similar curing conditions for both pozzolanic as well as for non-hydraulic mortars. For concrete and cement-based composites we use water curing.  Within the curing, the specimens were placed above the water reservoir that helped to maintain high relative humidity in curing box. However, the samples were placed on a steel grid in order to allow their carbonation. In this respect we believe, the curing conditions were ideal both for pozzolanic as well as for non-hydraulic mortar. Moreover, values of mechanical parameters obtained for 28 days samples were higher than considered typical for non-hydraulic mortars.

Point 7: Please, revise the mistake in the whole text and write Blaine instead of Blain.

We revised the whole text accordingly.

Point 8: Section 3.2, add XRD measurement conditions (angles, step size, …).

We added measurement conditions to the experimental part of the revised manuscript.

Point 9: Line 267, delete the number 6 in Figure 5, 6.

The text was revised accordingly.

Point 10: Line 223, revise the mistake and delete the final “e” in metakaoline.

We revised the above given mistake as requested.

Point 11: Line 342: please, write better “high pozzolanic character”.

The text was revised accordingly.

Reviewer 2 Report

The paper is aimed at experimentally investigating the suitability of binders, based on lime hydrate, metakaolin and biomass ashes to be used on producing eco-friendly lightweight mortars. The work contains outstanding materials and very complete (the authors did a huge experimental campaign covering chemical investigations, physical, thermal and mechanical properties). It is recommended its publishing.

In the Reviewer's opinion the paper needs few minor improvements before the approval.

The following recommendations/clarifications should be considered:

In section 1, after the State of the Art, the Authors should better clarify what are the key novelties of this paper and the main contributions of this work beyond the current SoA. They are missing or not deeply reported.

The water to binder ratios seem to be very high (even if the mortars will be employed for non-structural purposes).

For comparisons purposes I would expect to have comparisons between Mref and MBAM with the same w/b. Why did the author decide to cast composites with different w/b?

The authors should add a Table showing an overview of the Experimental Programme reporting type of test x maturation time x number of specimens x etc.

The number of specimens per each tests should be reported.

The scatter of the results should be also declared in all the results which were based on tests on different specimen repetitions (e.g. compression and flexural strengths, open porosity, transport properties and thermal ones).

Future developments which will follow to this research paper are only poorly outlined. It should be reported at the end of the concluding section.

Author Response

We highly appreciate work of reviewer 2 on review of our manuscript. We addressed all the reviewer’s comments and followed reviewer’s suggestions in the revised version of the manuscript. We believe that reviewer’s comments and suggestions have contributed substantially to the improvement of the presentation of our study and the overall quality of the manuscript. The revised text is in a revised manuscript marked up by a red colour.

Reviewer 2:

The paper is aimed at experimentally investigating the suitability of binders, based on lime hydrate, metakaolin and biomass ashes to be used on producing eco-friendly lightweight mortars. The work contains outstanding materials and very complete (the authors did a huge experimental campaign covering chemical investigations, physical, thermal and mechanical properties). It is recommended its publishing.

We thank reviewer 2 for such good evaluation of our paper and recommendation for paper publication.

The following recommendations/clarifications should be considered:

Point 1: In section 1, after the State of the Art, the Authors should better clarify what are the key novelties of this paper and the main contributions of this work beyond the current SoA. They are missing or not deeply reported.

As proposed by reviewer 2, we clarified the novelty of the paper. See the last paragraph of section Introduction.

Point 2: The water to binder ratios seem to be very high (even if the mortars will be employed for non-structural purposes).

We agree with reviewer 2, the w/b ratio was for both studied mortars high. It was due to the fact that we planned to developed mortar of high porosity what can be simply achieved by a higher amount of batch water. For commercial design of mortars based on reported ternary based binder, plasticizers enabling the reduction of batch water dosage can be used together with other chemical additives affecting for example mortar adhesiveness to substrates or its pore volume and pore size distribution. However, this was not subject of the submitted paper.

The text was modified accordingly.

Point 3: For comparisons purposes I would expect to have comparisons between Mref and MBAM with the same w/b. Why did the author decide to cast composites with different w/b?

We understand to this reviewer’s comment. However, based on our intensive research we made in the design of different types of construction materials, we followed different procedure in mortar mix design. In this case, the water/binder ratio was adjusted to get a similar workability of both mortars. It was verified by flow table tests. The value of spreading was 160/160 mm. As BA increased mortar workability, the water/binder ratio was decreased in MBAM mortar.

The text was modified accordingly.

Point 4:  The authors should add a Table showing an overview of the Experimental Programme reporting type of test x maturation time x number of specimens x etc.

As proposed by reviewer 2, we added Table 2 that gives an overview of the tests performed for hardened mortars samples. In this Table, samples maturation time and number of specimens used for the particular experimental test are also introduced.

Point 5: The number of specimens per each tests should be reported.

The number of specimens per arch test is newly introduced in Table 2.

Point 6: The scatter of the results should be also declared in all the results which were based on tests on different specimen repetitions (e.g. compression and flexural strengths, open porosity, transport properties and thermal ones).

Base on measurement uncertainty of particular test methods and obtained data, the scattering of the results was provided.

Point 7: Future developments which will follow to this research paper are only poorly outlined. It should be reported at the end of the concluding section.

As proposed by reviewer 2, the ideas of future applied research that will be necessary before launch the newly developed repair mortar in the market were outlined at the end of section 4. Conclusions.

Reviewer 3 Report

The manuscript focus on the development of the ternary cement binders for the possible application in lightweight mortars for the renovation purposes. The manuscript contains 14 pages in length together with 6 figures and 8 tables. The list of references is large and contains 55 positions. Even if the topic may be interesting, the manuscript lacks of systemathically planned research to see the effect of the different dosage of used binders. Due to this reson and the following ones I suggest to reject this manuscript at this stage.:

- the title is also not adequate to the content. I do not se any renovation aspect in this article. No attempt to renovate any structure using these moertars was conducted. What is the adhesion and compatibility of proposed mortars to the real structure or substrate?

- how the authors explain the increase of the compressive strength of mortars from 1.32 MPa for the reference one to 1.56 for MBAM while the porosity increase also fom 33% for the reference sample to 44% for MBAM sample? More detailed studies are needed to explain this behaviour,

- I would like to see more information in the introduction on the reason why lightweight cement mortars were developed? This is in the title and should be justified,

- Table. 1. It is assumed that the research was conducted based on just only two compositions of mortars (reference Mref and the second one called MBAM). Both mortars have the same binder/aggregate ratio but slightly different water/binder ratio (1 and 0.89),

- half of the results section (3 pages) is just the charactrization of used binders (biomass ash and commercially avalaible metakaolin and lime). The other half focus on the properties of these two mortars.

Author Response

We highly appreciate work of reviewer 3 on review of our manuscript. We addressed all the reviewer’s comments in the revised version of the manuscript. We believe that reviewer’s comments have contributed substantially to the improvement of the presentation of our study and the overall quality of the manuscript. The revised text is in a revised manuscript marked up by a red colour.

Reviewer 3:

The manuscript focus on the development of the ternary cement binders for the possible application in lightweight mortars for the renovation purposes. The manuscript contains 14 pages in length together with 6 figures and 8 tables. The list of references is large and contains 55 positions. Even if the topic may be interesting, the manuscript lacks of systemathically planned research to see the effect of the different dosage of used binders. Due to this reson and the following ones I suggest to reject this manuscript at this stage.

We are really sorry reviewer 3 doesn’t like our paper. We believe that our response to reviewer’s comments and significant improvement of manuscript in its revised version will change his opinion on our paper.

Point 1: The title is also not adequate to the content. I do not se any renovation aspect in this article. No attempt to renovate any structure using these moertars was conducted. What is the adhesion and compatibility of proposed mortars to the real structure or substrate?

We changed the paper title as proposed by reviewer 1. We believe that the renovation aspect of the paper and developed mortar is quite clear from the text what acknowledged also two other reviewers who positively evaluated idea of testing and developing new lime-based repair materials what is of interest for preserving architectural heritage.  Concerning adhesion of the developed mortar we agree with reviewer 3 that is an important parameter for mortar practical use. However, as stated in the manuscript, it was not subject of the paper at this stage of mortar design. The presented work was done to test possibility, whether the ternary blended binder with incorporated waste material coming from biomass combustion can be used for production of lime-pozzolan mortar for repair purposes. As the use of lime-pozzolan mortars is recommended and in most of cases even demanded by culture heritage authorities, the developed mortar brings beyond question alternative solution for renovation of historical buildings.  Moreover, the adhesion can be simply modified by chemical additives in production of final product that will be introduced in construction market. Future applied research will be conducted to solve all the problems with the application of the developed mortar in practice. At this stage of research, the mortar was designed and tested as universal base material that will be further modified according to the specific requirements for masonry and rendering mortars.

Point 2: How the authors explain the increase of the compressive strength of mortars from 1.32 MPa for the reference one to 1.56 for MBAM while the porosity increase also fom 33% for the reference sample to 44% for MBAM sample? More detailed studies are needed to explain this behaviour.

We agree with reviewer 3, the porosity affects mechanical parameters. However, in this case, the porosity was not only parameter contributing to the mechanical resistance of the developed ternary binder-based mortar. The increase in mechanical resistance we assigned to the formation of C-S-H and C-S-A-H products formed within the pozzolanic reaction of metakaolin and biomass ash with lime hydrate (see references [37-40]). This material performance is not surprising and was reported by many authors who aimed their research on improvement of lime mortars properties by the use of different types of pozzolans. Moreover, the high pozzolanic activity of BA and metakaolin was verified by Chapelle test.

The manuscript was revised accordingly.

Point 3: I would like to see more information in the introduction on the reason why lightweight cement mortars were developed? This is in the title and should be justified.

Thank you for this comment. The section Introduction and Conclusions were revised accordingly.

Point 4: Table. 1. It is assumed that the research was conducted based on just only two compositions of mortars (reference Mref and the second one called MBAM). Both mortars have the same binder/aggregate ratio but slightly different water/binder ratio (1 and 0.89).

The reported materials were designed based on out previous research conducted on cement-based composites with incorporated biomass ash and metakaolin. We think at this stage of research it was not necessary to introduce data for mortar mixes with different amount of ternary blended binder in mortar mix. We just wanted to verify, whether the developed binder will be applicable in lime mortar mix and what will be its contribution to the mortar structural, mechanical, hygric, and thermal performance.  Based on our previous studies dealing with application of pozzolans in cement and lime composites, similar trend in mechanical, hygric, and heat transport and storage parameters can be anticipated for different replacement ratio of lime hydrate.  In this paper we intended to present properties of well-workable mortar mix with the highest possible amount of BA in this respect. This was the main idea of our research and design process. Based on out previous results of pozzolanic activity of BA, we also tried to replace lime hydrate in mortar mix composition as much as possible, taking into consideration mechanical parameters of the developed mortar. As stated in section Conclusions, the studied topic demands future intensive research, where study of different dosage of binder constituents will be one of the research tasks. One must take into account fact, the paper aimed at completely novel application of a ternary blended binder based on BA, metakaolin and lime hydrate that was not studied up to now. The presented study can be therefore considered as first and initial attempt in this manner.  

The water/binder ratio was adjusted to get similar workability of both mortars. It was verified by flow table test. The value of spreading was 160/160 mm. As BA increased mortar workability, the water/binder ratio was decreased in MBAM mortar.

Point 5:- half of the results section (3 pages) is just the charactrization of used binders (biomass ash and commercially avalaible metakaolin and lime). The other half focus on the properties of these two mortars.

Yes, of course. Such paper organisation and presentation of research methods and obtained results are typical for studied field of materials research. First we performed detailed characterization of both pozzolans from the point of view of their chemical and physical properties, morphology, particle size distribution, etc. As this data were promising for their intended application in lime mortar mix, we further proceed to mortar mix design, samples preparation and curing. Finally, we conducted extensive experimental campaign and assessed studied mortars in respect to their structural, mechanical, hygric, and thermal properties. 

Round  2

Reviewer 1 Report

The authors have satisfactorily addressed all the queries raised in the first review report. They have included some new references and they have also explained the rationale behind the scientific work. They have provided a response concerning the final use of these mortars, either as structural (masonry) mortars or as renders. The authors have made a good improvement of the Experimental section, describing better methods and mix proportions. The use of BA as pozzolanic additive to improve the performance of lime-based mortars is of interest for repair works of the Architectural Heritage. Particular attention should be paid to the improvement in thermal conductivity properties: the reduction in thermal conductivity is an energy-saving mechanism for the repaired monuments. Even though some of these materials are now in a rather basic stage of the investigation, they appear as highly promising materials. The manuscript can be accepted for publication.

Author Response

We thank reviewer 1 for his comments and suggestions that significantly contributed to the improvement of the presentation of our study and the overall quality of the manuscript. As required by academic editors, we better underlined the goal of the paper especially in sections Abstract and Discussion.

Reviewer 3 Report

The changes are not significant. In my opinion mich work is required to obtain acceptable quality. Thus, I suggest to make changes and re-submit the article after substantial corrections

Author Response

We thank reviewer 3 for time he spent with review of our paper. We made required changes in first revision of our paper. We don’t agree with his opinion that the changes we made in our paper were not significant. Moreover, we addressed all his comments as suggested in first revision and in the second review round, he didn’t formulate any exact lack of the paper.